# Molecular Pro-Apoptotic Activities of Flavanone Derivatives in Cyclodextrin Complexes: New Implications for Anticancer Therapy

**DOI:** 10.3390/ijms25158488

**Published:** 2024-08-03

**Authors:** Angelika A. Adamus-Grabicka, Pawel Hikisz, Artur Stepniak, Magdalena Malecka, Piotr Paneth, Joanna Sikora, Elzbieta Budzisz

**Affiliations:** 1Department of Bioinorganic Chemistry, Faculty of Pharmacy, Medical University of Lodz, Muszynskiego 1, 90-151 Lodz, Poland; joanna.sikora@umed.lodz.pl; 2Department of Molecular Biophysics, Faculty of Biology and Environmental Protection, University of Lodz, Pomorska 141/143, 90-236 Lodz, Poland; pawel.hikisz@biol.uni.lodz.pl; 3Department of Physical Chemistry, Faculty of Chemistry, University of Lodz, Pomorska 163/165, 90-236 Lodz, Poland; artur.stepniak@chemia.uni.lodz.pl (A.S.); magdalena.malecka@chemia.uni.lodz.pl (M.M.); 4Institute of Applied Radiation Chemistry, Lodz University of Technology, Zeromskiego 116, 90-924 Lodz, Poland; piotr.paneth@p.lodz.pl; 5Department of the Chemistry of Cosmetic Raw Materials, Medical University of Lodz, 90-151 Lodz, Poland

**Keywords:** anticancer properties, biocompatibility, crystal structure, cyclodextrines, docking studies, flavonoid derivatives, inclusion complexes

## Abstract

This study evaluates the antiproliferative potential of flavanones, chromanones and their spiro-1-pyrazoline derivatives as well as their inclusion complexes. The main goal was to determine the biological basis of molecular pro-apoptotic activities and the participation of reactive oxygen species (ROS) in shaping the cytotoxic properties of the tested conjugates. For this purpose, changes in mitochondrial potential and the necrotic/apoptotic cell fraction were analyzed. Testing with specific fluorescent probes found that ROS generation had a significant contribution to the biological anticancer activity of complexes of flavanone analogues. TT (thrombin time), PT (prothrombin time) and APTT (activated partial tromboplastin time) were used to evaluate the influence of the compounds on the extrinsic and intrinsic coagulation pathway. Hemolysis assays and microscopy studies were conducted to determine the effect of the compounds on RBCs.

## 1. Introduction

The rapid increase in life expectancy has been accompanied by a greater risk of various diseases. One of the most important and difficult challenges faced by modern science is the presence of cancer. According to data provided by the World Health Organization (WHO), approximately one in five people struggle with cancer [1,2].

In recent years, heterocyclic compounds of natural origin have played a very important role in the design of anticancer drugs, both in their original and chemically modified forms. Indeed, one of the most effective ways of identifying new anticancer drugs is through the chemical modification of naturally occurring substances with proven high anticancer activity. More than 65% of anticancer drugs approved by the Food and Drug Administration (FDA) between 2010 and 2015 contained a heterocyclic ring, which has a particularly important role in drug design [3]. Most heterocycles in available anticancer drugs contain a nitrogen atom in their structure, as this increases their potential to induce the death of cancer cells [4]. However, these compounds may be characterized by low solubility in water, which lowers their bioavailability and hence activity in the body. To address this, the solubility and bioavailability of the incorporated compound can be increased by synthesizing cyclodextrin inclusion complexes.

Cyclodextrins (CDs) are a group of cyclic oligosaccharides, among which the three best known and of greatest practical importance are α-, β- and γ-cyclodextrin. The molecules of these chemical compounds are composed of six, seven and eight glucose units, respectively, connected by α-1,4-glycosidic bonds [5]. Cyclodextrins have been well known in the food and pharmaceutical industries for several decades [6,7]. Over this time, it was found that CDs have the ability to form inclusion complexes that allows them to dissolve and stabilize drugs [8,9,10]. Such dissolution improves the chemical and pharmacokinetic properties of the compound and allows the formulation of appropriate drug forms. Due to their hydrophilic surface, the CD matrix has relatively low affinity for lipophilic biological membranes; as such, the molecules remain on the surface, while the hydrophobic drug molecules placed in their gaps penetrate the membranes [11]. Therefore, they are used to increase the penetration of the drug substance through biological membranes and improve its bioavailability. As such, CDs have proven valuable in various drug forms, such as dermatological preparations, mouthwashes and eye and nose drops [12,13].

For many years, our team has conducted research on the synthesis of biologically active compounds that contain a heterocyclic ring bearing an oxygen atom (chromanone/flavanone derivatives) and nitrogen atoms (spiro-1-pyrazolines) [14]. The biological activity, bioavailability and physicochemical properties of the synthesized compounds have been tested using quantum chemical calculations [15,16,17,18]. These studies have also compared the structures of a series of flavonoid and spiro-1-pyrazoline derivatives with their cytotoxic activity [19]. Further research has also examined the biological activity of compounds with a methyl group at the meta and para positions of the phenyl ring attached to the heterocyclic pyrazoline ring of spiro-1-pyrazoline [20].

The greatest obstacle regarding the use of the tested compounds as drugs concerns their solubility in aqueous solution. Therefore, the aim of the present study is to synthesize inclusion complexes with selected cyclodextrins with higher solubility. The study also examines the thermodynamics of the complexation process between the tested compounds and the cyclodextrins and subjects them to docking studies and a biological evaluation.

Among the many synthesized compounds presented in previous articles, those that showed the best cytotoxic activity against selected cancer cell lines were chosen for inclusion the present paper: **1**, **2**, **3**, **3a**, **4** and **5** (Figure 1). Of these, three compounds, **3**, **3a** and **5**, were used for the synthesis of inclusion complexes with cyclodextrins: these compounds demonstrated a methoxy group located in the *meta* position (**3**) and lacked any substituent in the phenyl ring (**5**). In previous studies, these compounds have demonstrated high cytotoxic activity against cancer cell lines. The compounds were complexed into the cyclodextrins and their antiproliferative properties were tested again.

Following this, two pairs of compounds were selected to determine the clotting time: **3** and **β-CD + 3**, **5** and **β-CD + 5**. The reference compound was β-CD.

## 2. Results and Discussion

### 2.1. Biological Assay

#### 2.1.1. Determination of Cytotoxicity of Chromanone Analogues Condensed with Pyrazolines by Metabolic Microplate Spectrophotometric Assay with MTT

The antiproliferative properties of the six tested chromanone derivatives (compound **1**, **2**, **3**, **3a**, **4** and **5**) were tested in breast cancer cells (MCF-7, MDA-MB-231 and HCC38) and endometrial cancer cells (Ishikawa, Hec-1-A) using the MTT test. Based on the obtained cell survival results, the IC_50_ concentration was determined for each compound. Based on the results, the tested pyrazoline derivatives were divided into two groups: one with high antiproliferative activity against all tested cancer lines and another with clearly weaker anticancer activity. Interestingly, all the compounds tested showed quite strong polarization of biological anticancer activity: either they had a very satisfactory effect on the tested cancer cells and their IC_50_ concentration was often close, or even slightly lower, to the reference compound (cisplatin), or their activity was very low regardless of the cancer lines.

The most active compounds were derivatives **1**, **3** and **5**. Interestingly, no significant differences in IC_50_ values were noted between estrogen-dependent and estrogen-independent breast cancer lines, which may suggest that the biological anticancer activity of the derivatives is not influenced by the presence of the estrogen receptor. Regarding the endometrial line, the Ishikawa line, was slightly more sensitive to the tested compounds: the IC_50_ concentrations for compounds **1**, **3** and **5** were approximately 10 µM lower than those for HMEC-1.

Compound **1** demonstrated the greatest antiproliferative effect based on IC_50_ values (Table 1). The IC_50_ value ranged from approximately 10 to 18 µM depending on the cancer line. Compound **3** had a slightly worse biological effect, with an IC_50_ of 15–27 µM. The least active compound (**5**) had an IC_50_ of 18–36 µM. The remaining compounds, viz., **2**, **3a** and **4**, demonstrated lower biological activity against all cancer cells. In the vast majority of cases, in the lower part of the concentration range used (10–120 µM), no significant decrease in tumor cell proliferation was observed compared to cisplatin.

The analysis of the cytotoxicity results obtained from the MTT method allows us to notice differences in the biological cytotoxic and antiproliferative activity of the six tested flavanone derivatives. Interestingly, a certain tendency in biological activity was always maintained for each of the analyzed compounds. Individual compounds were either characterized by very attractive antiproliferative and cytotoxic properties, with an IC50 similar to or even better than the reference compound cisplatin, regardless of the cancer cell line used (compounds **1**, **3** and **5**), or they showed very weak cytotoxic activity (compounds **2**, **3a** and **4**) in the range of 10–200 µM for all six cancer lines. The obtained results allow us to hypothesize that the total anticancer activity of individual flavanone derivatives is primarily influenced by the structure–activity relationship (SAR). Changes in the structure of a molecule may lead to changes in its ability to interact with receptors or other biological molecules, which in turn may affect its biological activity.

Interestingly, the most active derivatives in their chemical structure at the C3 position of the benzene ring did not have a p,o-methylophenyl-4,5-dihydro-3H-pyrazole ring attached but a pi double bond connecting another aromatic benzene ring. At the same time, the attachment at the C3 position of five-membered (pyrazoles) ring with a double bond between nitrogen atoms resulted in a significant reduction in the antiproliferative activity of the analyzed derivatives. It should be noted that such a structure–biological activity relationship was always characteristic of all the tested molecules. Therefore, it seems likely that this structural change may be crucial in the context of overall anticancer activity against the cancer cell lines used in the studies.

Moreover, in relation to our research, there is a clear dependence of the antiproliferative properties of the tested compounds on their pro-oxidant activity. The most biologically active compounds were able to generate ROS in cancer cells. The issue of the level of ROS in cancer cells is discussed in more detail in Section 2.1.6.

Based on the obtained MTT results and IC_50_ values, the most biologically active pyrazoline derivatives were chosen for further research: only compounds with an IC_50_ of 40 µM or below were selected. Hence, compounds **1**, **3** and **5** were taken for further detailed biological studies. The selectivity of their biological activity was determined against the HMEC-1 immortalized human microvascular endothelial cell line using the MTT test. The results (Table 1) highlight the important problem of the selectivity of potential chemotherapeutics and side effects of chemotherapy. The calculated IC_50_ values indicate that although these compounds have a weaker effect on normal cells than cancer cells, they still cause cell death: the calculated IC_50_ values were approximately 10 µM higher for the HMEC-1 line than the cancer lines, but they still remained relatively high, i.e., approximately 26–30 µM.

#### 2.1.2. Determination of Cytotoxicity of Chromanone Analogues Condensed with Pyrazolines Incorporated in Cyclodextrins by Metabolic Microplate Spectrophotometric Assay with MTT

An additional aspect of our research was the synthesis of chromanone analogues condensed with pyrazolines incorporated in cyclodextrins and the preliminary assessment of their anticancer activity. For this purpose, guest-host inclusion complexes with cyclodextrins (CD) were created for compounds **3**, **3a** and **5**. Their cytotoxicity towards the HMEC-1 line and anticancer activity against MCF-7, MDA-MB-231 and HCC38, Ishikawa and Hec-1-A was also tested using the classic MTT test.

Pure cyclodextrin was not found to demonstrate any cytotoxic activity towards HMEC-1 endothelial cells or any tested cancer cells in the entire range of tested concentrations (10–120 µM). The inclusion of the pyrazoline derivatives in cyclodextrin generally resulted in a total loss of anticancer activity in the tested concentration range. The non-included derivatives **3** and **5** were characterized by high biological activity and effectively inhibited the proliferation of breast and endometrial cancer cells; however, their inclusion in cyclodextrins resulted in the complete inhibition of anticancer activity. It should, however, be emphasized that the encapsulation of pyrazoline derivatives in cyclodextrins altered their cytotoxicity towards HMEC-1 and resulted in a significant seven- to eight-fold increase in IC_50_ compared to the unincorporated form. The results of the MTT test for pyrazoline derivatives incorporated in CD are presented in Table 2.

#### 2.1.3. Analysis of Changes in Transmembrane Mitochondrial Potential (ΔΨm)—JC-1 Method

Mitochondria play an important role in the process of cell death, both related to the formation of rapid necrotic changes (necrosis) and apoptotic pathways. These organelles play a key role in the course of programmed cell death (PCD), mainly by activating the apoptosis pathway in the internal (mitochondrial) pathway. This can be caused by an increase in mitochondrial membrane permeability (MMP) due to, among others, the overproduction of ROS, or overload of the organelles with Ca^2+^ ions due to their increased uptake from the cytoplasm. It is believed that the beginning of irreversible cell death is determined by two interconnected molecular events: changes in the permeability of the mitochondrial membrane (depolarization/hyperpolarization of the membrane potential) and the activation of caspases.

The changes in mitochondrial potential (ΔΨm) occurring in cancer cells exposed to the tested compounds were determined by fluorescence with the JC-1 probe. The results are presented in Figure 1a. In the case of breast cancer tumor lines (MCF-7, MDA-MB-231 and HCC38), changes in mitochondrial potential were observed in all mentioned cell lines for all tested compounds. Interestingly, for the MCF-7 line, only compound **1** caused depolarization of the mitochondrial potential (a decrease of approximately 17% in ΔΨm compared to the control), while a clear, very strong hyperpolarization was observed in the case of compounds **3** and **5** by approximately 30 and 40%, respectively. Twenty-four hour incubation with the tested pyrazoline derivatives resulted in a significant decrease in ΔΨm in MDA-MB-231 and HCC38 cells ranging from approximately 20 to 25%. The only exception here is the HCC38 line treated with compound **5**; in this case, similar to MCF-7, an increase in mitochondrial potential was observed, as well as strong hyperpolarization.

The endometrial cancer cells (Ishikawa and Hec-1-A) demonstrated similar changes in ΔΨm to the breast cancer cells. It should be emphasized, however, that in Hec-1-A, statistically significant changes in mitochondrial potential, i.e., approximately 30% hyperpolarization compared to control, were observed only for compound **1**: no changes in ΔΨm were observed for derivatives **3** and **5**. The Ishikawa cultures also demonstrated dysfunctional mitochondria, indicating the initiation of apoptosis. A particularly high decrease in ΔΨm was observed for compound **1** (about 30%) and a slightly smaller decrease for compound **3** (about 15%). Interestingly, as in the case of the MCF-7 and HCC38 breast cancer cell lines, exposure of the Ishikawa line to derivative **5** resulted in strong hyperpolarization of mitochondrial membranes (approximately 30% relative to controls).

The changes in mitochondrial potential accompanying the induction of apoptosis were confirmed by simultaneous fluorescence microscopy imaging and spectrofluorimetric measurements (JC-1 probe). In healthy cells with an intact mitochondrial membrane, i.e., not involved in the apoptosis pathway, the JC-1 dye is stored in the form of fluorescent aggregates with red-orange fluorescence. During depolarization of the mitochondrial membrane, or in apoptotic cells, JC-1 remains in the form of monomers that exhibit green fluorescence. The cancer cells exposed to the test compounds showed strong green fluorescence (Figure 1b), indicating damage to the mitochondrial membranes.

#### 2.1.4. Changes in Plasma Membrane Fluidity (Measurement of TMA-DPH and DAUDA Fluorescence Anisotropy)—Lipid Peroxidation

ROS can influence the fate of the cell by inducing lipid peroxidation. This is an ROS-dependent oxidation process, primarily of polyunsaturated fatty acids that are part of phospholipids. This process is free radical in nature, and cells exposed to oxidative stress experience increasingly intense peroxidation reactions. The process disturbs the integrity of the cell membrane, thus increasing free oxygen radical production in the cell and ultimately cell death.

The changes in the fluidity of cancer cell membranes were determined using fluorescence anisotropy measurements of two fluorescent probes located at different depths of the lipid bilayer: DAUDA, which can be used to indicate fluidity around deep fatty acids (hydrophobic), and TMA-DPH, located on the hydrophilic heads of membrane phospholipids.

This analysis indicated that the test compounds caused diverse changes in the structure of the lipid bilayer (Figure 2 and Figure 3). The increase, or decrease, in membrane fluidity appears to be correlated with the type of cancer cell; however, the data suggests that the only significant changes in the fluidity took place in the deeper, hydrophobic layers. It should be noted that the only clear increase in membrane fluidity in the hydrophilic area (TMA-DPH probe) occurred in the HCC38 breast cancer line; in this case, fluidity increased by approximately 45–55% compared to controls for the three analyzed derivatives. In the case of the remaining four cancer cell lines, no significant changes in lipid peroxidation in the hydrophilic surface region were generally observed; however, the Hec-1-A and Ishikawa lines treated with compounds **3** and **5**, respectively, demonstrated an approximately 20% increase in cell membrane fluidity.

Significantly greater changes in fluidity were observed in the hydrophobic layers (DAUDA probe) for all cancer cell lines. Interestingly, incubation with the tested derivatives resulted in decreased membrane fluidity (25–45% compared to controls) in the breast cancer lines (MCF-7, MDA-MB-231 and HCC38) but increased fluidity in the endometrial lines (Ishikawa and Hec-1-A).

The observed changes in membrane fluidity (Figure 2 and Figure 3) confirm that the analyzed chromanone derivatives had an influence on the lipid bilayer. Moreover, the correlation with ROS generation suggests that free radicals may play a key role in the peroxidation of membrane lipids.

#### 2.1.5. Determination of the Fraction of Apoptotic and Necrotic Cells by Fluorescence Microscopy (Double Staining of Cells with Hoechst 33258 and Propidium Iodide Fluorescent Dyes

To determine the type of cell death induced by the investigated compounds in breast and endometrium cancer cells, the cells were subjected to Hoechst 33258/propidium iodide staining, and the morphological changes were analyzed. Generally, all pyrazoline derivatives induced primarily apoptosis in cancer cells after 24 h incubation; importantly, the percentage of apoptotic cells (bright blue fluorescence) was significantly higher (approximately 20–25%) than necrotic cells (approximately 10–15%) (red fluorescence). It is worth emphasizing that slightly more necrosis appeared in the case of breast cancer cells than in the case of endometrial cells. Interestingly, after 24 h of incubation, only small numbers of late apoptotic cells (violet-blue fluorescence) were noted, amounting to only about 2–3% of cells compared to the control. The results are presented in Figure 4.

Double-staining allowed for the morphological analysis of the treated cells. Typical changes associated with apoptosis, such as highly condensed chromatin within a shrunken nucleus, were observed in all cells treated with the pyrazoline derivatives. The derivatives also caused visible cell shrinkage. The observed morphological changes were connected with the induction of apoptosis. Hence, it appears that apoptosis and necrosis may contribute to the cytotoxic and antiproliferative effects of the investigated pyrazoline derivatives in breast and endometrial cancer cells. Examples of the typical changes observed for these compounds are shown in Figure 5.

#### 2.1.6. Analysis of the Generation of Reactive Oxygen Species (ROS) and Reactive Nitrogen Species (RNS)

Reactive oxygen and nitrogen species generation in the treated breast and endometrial cancer cells was measured using specialized fluorescent probes. In the case of ROS, two probes were used: DCFDA (Figure 6) for superoxide anions and H_2_DCFDA (Figure 7) for hydrogen peroxide. In turn, reactive nitrogen species were assessed by a DAF-FM (Figure 8) diacetate probe, which can detect and quantify low concentrations of nitric oxide.

Incubation of cancer cells led to the generation of free radicals. The level of ROS produced varied depending on the cell line and fluorescent probe. The greatest changes in superoxide anion (DHE probe) growth were observed for compounds **3** and **5**; in the MCF-7, MDA-MB-231 and Ishikawa lines, treatment resulted in approximately 20% (**3**) and 30% (**5**) greater superoxide anion levels relative to control. They also induced the formation of 10–15% ROS in the HCC38 line. In the case of compound **1**, the only significant increase in radical anion was observed for the MDA-MB-231 and HCC38 lines, this being approximately 20% compared to control cells. Interestingly, in Hec-1-A cells, superoxide anion levels rose by approximately 20% compared to the control after incubation but only for compound **3**.

An increase in hydrogen peroxide in cancer cells was also observed after 24 h of incubation. However, the kinetics of these changes were slightly different than those observed for the superoxide anion radical. A more intense reaction was observed for compound **1**, yielding an approximate 15–25% increase in all lines except HCC38. The MCF-7 and MDA-MB-231 lines were particularly sensitive, demonstrating a 20–40% increase; among these, a particularly strong pro-oxidant effect was demonstrated for compound **3**, with an approximate 30–40% increase in H_2_O_2_ concentration. Interestingly, as noted for the superoxide anion, the DHE probe, endometrial cancer lines were also slightly less sensitive to the pro-oxidant effects of treatment, with a 15–20% reduction in H_2_O_2_ level observed for all analyzed derivatives compared to the breast cancer lines.

Unlike ROS, treatment resulted in a lesser increase in reactive nitrogen species (RNS). In the case of the MCF-7 and Ishikawa lines, no statistically significant changes in RNS were observed after incubation with any pyrazoline derivative. In the case of HCC38 and Hec-1-A, only compound **1** resulted in an increase in nitric oxide production, with an approximate increase of 15% and 20%, respectively. This is an interesting result, since compound **1** demonstrated the lowest ROS generation among the three treatments. Indeed, incubation of compound **1** with MDA-MB-231 line increased nitric oxide level by approximately 20%. Additionally, this line remained the most sensitive to the pyrazoline-dependent generation of RNS, as compounds **3** and **5** also induced an approximately 15% increase in nitrogen radical production.

One of the chemotherapy strategies is to increase oxidative stress in cancer cells by either indirectly promoting the generation of reactive oxygen species or directly inducing their formation, leading to the overproduction of ROS [21]. The production of ROS in cancer cells leads to the death of cancer cells by apoptosis. Several studies have reported reduced antioxidant status with increased ROS levels in oncology patients undergoing treatment [22,23].

The results of our research indicate that the analyzed derivatives induce oxidative stress in cancer cells, contributing to an increase in the level of reactive oxygen species, both superoxide anion O_2_^•−^ and hydrogen peroxide H_2_O_2_. Moreover, MTT tests conducted using antioxidants proved that the role of the generated ROS in shaping the total antiproliferative potential of the tested flavanone derivatives is significant. Antioxidants effectively limited the cytotoxic activity of the analyzed conjugates. Importantly, the pro-oxidant properties of flavanone derivatives are closely related to their ability to inhibit the activity and reduce the level of glutathione (GSH) in cancer cells. This enzyme is a key intracellular antioxidant that is able to directly neutralize superoxide anion radicals. In our other studies conducted on colon cancer lines (results not yet published), derivatives **1**, **3** and **5** caused a decrease in GSH concentration in cancer cells. As our results indicate, the reduction of GSH levels in cancer cells dependent on the tested derivatives is important because this protein may determine the effectiveness of chemotherapy. It has been proven that high GSH levels are independently associated with resistance to chemotherapy and radiotherapy [24]. In vitro studies indicate that intracellular GSH deficiency makes cancer cells more sensitive to oxidative stress, overcoming drug resistance and further improving the results of anticancer therapy [25,26].

An important issue requiring further research is how the analyzed flavonoid derivatives generate ROS in cancer cells. There is evidence that polyphenols, despite being antioxidants, can also have anti-cancer properties by creating oxidative stress in cancer cells [27]. We assume that the pi double bond systems found in the structure of the tested flavanone derivatives are crucial for generating ROS. Their susceptibility to breakdown makes them good candidates for reactions that generate free radicals. Additionally, pi double bonds can act as catalysts and attachment sites for free radicals, which can lead to further generation of free radicals. Pi double bonds can act as catalysts for reactions generating free radicals through autoxidation. The substrate of these reactions is molecular oxygen with which flavanone derivatives react. The product of a one- or two-step electron transfer reaction is the production of superoxide anion radicals. In the next step, certain parts of flavonoid molecules (hydrogen atom) can react with superoxide radicals. In this process, the flavonoids donate a hydrogen atom, leading to the formation of hydrogen peroxide (H_2_O_2_). Moreover, studies have shown that a carbonyl group at position 4 of the C-ring all influence how effectively flavonoids act as pro-oxidants [28].

The next step in generating ROS, depending on the tested flavanone derivatives, may be the generation of a highly reactive hydroxyl radical. The resulting hydrogen peroxide, mainly with the participation of Fe^2+^ iron ions, undergoes the Fenton reaction with the final formation of a hydroxyl radical. This radical, as one of the most active free radical molecules, significantly accelerates DNA damage and the formation of 8-oxo-2′-deoxyguanosine (8-oxo-dG), the presence of which results in DNA mutations and eventual cell death.

We emphasize that this is a proposed hypothetical model of the way in which the tested derivatives generate free radicals in cancer cells. It is significant that, in the case of derivatives with very low cytotoxic properties (**2**, **3a**, **4**), no statistically significant changes in the level of reactive oxygen species generation were observed (results not published). It should be noted that the effect of pi double bonds on the generation of free radicals depends on many factors, such as the type of pi double bond, the surrounding environment, and the presence of other chemicals. Undoubtedly, in the future, we plan to further research this area in the field of molecular biology, which will shed a little more light on the discussed issue.

#### 2.1.7. Assessment of the Contribution of Reactive Oxygen Species to the Cytotoxicity of the Cytotoxic Activity of Chromanone Derivatives

The impact of ROS on the cytotoxic activity of pyrazoline derivatives was determined in breast and endometrial cancer cells. The cells were first incubated for one hour with antioxidants, viz., N-acetylcysteine (NAC) or a water-soluble derivative of vitamin E (Trolox) and were then cultured with the test compounds.

Preincubation of cancer cells with antioxidants caused a significant decrease in the cytotoxic properties of chromanone derivatives, which was reflected in an increase in the IC_50_ value. Figure 9 shows the IC_50_ values of pyrazoline derivatives obtained in the cancer cells, either with or without pre-incubation with NAC antioxidants or Trolox. The pre-incubated variants demonstrated a statistically significant increase in IC_50_ value for all derivatives used. It is worth noting, however, that in most variants of the experiment, NAC was more effective than Trolox as a scavenger of free radicals generated by the tested compounds. These results (Table 3) indicate that the pro-oxidant properties of the analyzed chromanone derivatives, and their potential to generate ROS, play key roles in their biological anticancer activity.

#### 2.1.8. The Results of the Biocompatibility Assessments

The application of erythrocytes as a model for cytotoxicity screening of xenobiotics or new compounds is common in biomedical science. Morphological changes, subsequent disruption of RBC membrane integrity and hemolysis can be used to determine the potential cytotoxicity of various compounds [29]. The effect of the tested compounds in the concentration range of 1–100 µmol/L on damage to the protein-lipid membrane of RBCs, expressed as % hemolysis, is presented in the Appendix A. A 24 h incubation of RBCs with β-CD did not have any adverse effect on the erythrocyte membrane, and the recorded % hemolysis values did not significantly differ from the control sample containing saline solution, corresponding to spontaneous hemolysis. This indicates the hemocompatibility of β-CD. This is further confirmed by microscopic images of erythrocytes incubated with β-CD (Figure 10), where no pathological changes in RBCs or damaged cells were observed. Interestingly, after incubation with β-CD, the occurrence of echinocytes and stomatocytes was observed at all tested concentrations. This is a physiological and reversible phenomenon, indicating the permeation of the compounds through the protein-lipid membrane.

In contrast, compounds **3** and **5** and their complexes with β-CD did not show a statistically significant impact within the concentration range of 1–25 µmol/L. However, at concentrations of 50 µmol/L and 100 µmol/L, there was a statistically significant increase in % hemolysis. Therefore, the compounds can only be considered biocompatible and not damaging to the cell membrane up to a concentration of 25 µmol/L.

In the morphological images of RBCs (Figure 10), after incubation with the studied compounds, single echinocytes and stomatocytes were observed, which is a physiological and reversible phenomenon, which confirmed the permeation of the compounds through the protein-lipid membrane. In the case of incubation with 100 µmol/L of compound **3**, the appearance of single eryptocytes was observed, which is a form of programmed RBC death. Eryptosis could be triggered by several factors, including an osmotic shock or energy depletion in erythrocytes, ROS formation and depletion of antioxidants.

None of the studied compounds showed a statistically significant impact on the extrinsic coagulation pathway (Figure 11), expressed as prothrombin time (PT). Regarding the intrinsic coagulation pathway (Figure 12), incubation with compound **3**, as well as **5** and **β-CD + 5**, resulted in a mild, statistically significant prolongation of APTT time; however, these values did not exceed the reference range for this parameter. The mild anticoagulant effect of the compounds with potential anticancer activity may be beneficial, since both breast and uterine cancers are associated with an increased risk of thromboembolic events.

The TT parameter (Figure 13), indicating the rate of conversion of fibrinogen to fibrin under the influence of exogenous thrombin, showed that the shortening of TT time observed in the case of the studied compounds did not exceed the reference range for this parameter and did not exhibit a prothrombotic effect.

### 2.2. Studies of Physicochemical Measurements

#### 2.2.1. ITC

In order to determine the thermodynamics of the complexation process of the tested ligands in the hydrophobic interior of cyclodextrin, isothermal calorimetric titration measurements were performed. The thermal effect that describes the direct interaction of **3**, **3a** and **5** with α-CD, β-CD and HP-β-CD in DMSO solutions as a function of the composition of the titrated solution is shown in Figure 14, Figure 15 and Figure 16. A single active site model was used to mathematically describe the obtained thermograms. Based on the model, the stoichiometry of the resulting inclusion complex (*n*), its formation constant (K), molar enthalpy (ΔH) and entropy (ΔS) of the complexation process, and the Gibbs free energy (ΔG) were determined. The values of the thermodynamic parameters that describe the interaction of the tested ligands with α-CD, β-CD and HP-β-CD are listed in Table 4, Table 5 and Table 6.

The stoichiometric ratio (*n*), i.e., the number of ligand molecules per macromolecule of cyclodextrin, was determined calorimetrically; the result was close to one for all compounds tested (Table 4, Table 5 and Table 6). This indicates the formation of complexes with a stoichiometry of 1 (CD): 1 (compounds). The complexation process of the compounds tested was exothermic (ΔH < 0). For alpha-cyclodextrin, the enthalpy of the reaction with all tested ligands is much lower than that for β-CD and HP-β-CD containing seven glucose molecules in their rings. This is probably due to a poorer spatial fit to the smallest cyclodextrin, which is also confirmed by ACD having the lowest value (*n*) for the complexes. All complexation processes were accompanied by an increase in the degree of disorder of the reactants (ΔS > 0). Compounds **3**, **3a** and **5** all demonstrated spontaneous binding to the tested cyclodextrins (ΔG < 0).

The ongoing process had a spontaneous nature, as indicated by the high complexation constants (logK > 3) for the inclusion complexes [30], as noted previously [31,32]. The calculated complexation constant increased along the series α-CD < β-CD < HP-β-CD. The greater stability of β-CD compared to α-CD can be attributed to the better spatial fit of the ligand molecule to the hydrophobic gap of the macrocycle, with this resulting from the larger internal diameter of the β-CD molecule [33]. HP-β-CD had a higher complexation constant than β-CD; this is probably due to additional interactions associated with the ligand molecule incorporated into the hydrophobic cavity of the macrocycle with the hydroxypropyl groups of HP-β-CD [34].

#### 2.2.2. UV–Vis

The effect of cyclodextrin concentration on the water solubility of compounds **3**, **3a** and **5** in water was confirmed by UV–Vis spectrophotometry (Figure 17, Figure 18 and Figure 19). The obtained solubility diagrams can be classified as type A_L_ based on Higuchi and Connors [35]. Hence, cyclodextrin incorporation was associated with greater water solubility of the ligand. The solubility graphs show a linear relationship for all tested cyclodextrins (Figure 17, Figure 18 and Figure 19). Furthermore, the addition of cyclodextrin to an aqueous solution of the ligand appears to increase its solubility in water (Table 7).

At a concentration of 15 mM (i.e., the maximum solubility of β-CD), compounds **3**, **3a** and **5** were more soluble in water in the presence of β-CD than α-CD. This is due to the larger β-CD molecule (seven glucose molecules) having a better spatial fit to the shape of the ligand molecule [36]; also up to 15 mM, compounds **3**, **3a** and **5** with modified β-cyclodextrin demonstrated a similar increase in solubility as the unsubstituted β-CD. The solubility of the tested ligands continued to increase above 15 mM HP-β-CD. At the maximum concentration analyzed for HP-β-CD (90 mM), the solubility of the tested ligands in water increased from 28 to 132 times (Table 7).

### 2.3. Docking Studies

Docking scores are collected in Table 8 where columns correspond to structures of cyclodextrins and rows to structures of ligands. As can be seen, the differences are not very large, with the strongest interactions being observed for spiro ligands docked to LEDROB [37] dextrin, and the weakest for non-spiro ligands docked to α-CD. The strongest complex is illustrated in Figure 20. As no hydrogen bonds or stacking interactions have been identified, presumably, the stability of the complexes is controlled solely by electrostatic interactions.

## 3. Materials and Methods

### 3.1. Biological Assay

Solutions of the tested compounds were prepared by dissolving the sample in DMSO to obtain a concentration of 5 mmol/L. β-CD inclusion complexes with ligands were prepared at a concentration of 5 mmol/L as follows. β-CD was dissolved in water and stirred without temperature compensation on a magnetic stirrer. Ligand solutions were slowly added dropwise and left for 24 h on a magnetic stirrer. The solutions were then placed in a refrigerator to establish dynamic equilibrium. The solutions ready for testing were diluted to the appropriate concentrations: 120 µmol/L, 100 µmol/L, 50 µmol/L, 30 µmol/L, 20 µmol/L and 10 µmol/L.

#### 3.1.1. Biological Material, Culture and Passage of Cells

Biological tests were performed on three breast epithelial adenocarcinoma cell lines: one positive for estrogen and progesterone receptors MCF-7 (ATCC^®^ HTB-22™), and two negatives for estrogen and progesterone receptors MDA-MB-231 (ATCC^®^ HTB-26™) and HCC38 (ATCC^®^ CRL-2314™). These were accompanied by two endometrial cancer lines: Ishikawa (99040201; Sigma-Aldrich, St. Louis, MO, USA) and Hec-1-A (ATCC^®^ HTB-112™). In addition, human microvascular endothelial cell culture HMEC-1 (obtained from the American Type Culture Collection, Manassas, VA, USA) was used as a model of normal cells to assess the cytotoxicity of the analyzed pyrazoline derivatives.

HMEC-1 cells were cultured in MCDB 131 (Corning Life Sciences, Corning, NY, USA) culture medium supplemented with 10% fetal bovine serum (Gibco, Grand Island, NY, USA) and antibiotics (10 U/mL penicillin and 0.5 mg/mL streptomycin) (Gibco, Grand Island, NY, USA). The culture medium was enriched with L-glutamine (10 M), hydrocortisone (1 μg/mL) and epidermal growth factor EGF (10 ng/mL) (Millipore, Burlington, MA, USA). MCF-7, MDA-MB-231, Ishikawa and Hec-1-A tumor cells were cultured in Dulbecco’s minimal essential medium (DMEM, Lonza, Visp, Switzerland) culture medium supplemented with 10% fetal bovine serum and antibiotics (10 U/mL penicillin and 0.5 mg/mL streptomycin). HCC38 tumor cells were cultured in RPMI (Thermo Fisher Scientific, Waltham, MA, USA) culture medium supplemented with 10% fetal bovine serum and antibiotics (10 U/mL penicillin and 0.5 mg/mL streptomycin).

The cultures were carried out in a CO_2_ cell culture incubator at a temperature of 37 °C in an atmosphere of 5% CO_2_ and relative humidity of 100%. Cells were maintained in the logarithmic growth phase by regular passaging 2–3 times a week with 0.25% trypsin-EDTA (Gibco, Grand Island, NY, USA). After removing the culture medium, the cell monolayer was washed with 0.9% NaCl, and 300–500 μL of 0.25% trypsin with EDTA solution was added: the amount of trypsin depended on the surface of the culture vessel. The vessel/dish was placed in a CO_2_ incubator for 3–5 min. The trypsinization process was monitored under a microscope. After its completion, trypsin was gently removed, and culture medium was added at an appropriate volume for a given culture vessel. After mixing thoroughly, the cell suspension was transferred into new, sterile culture vessels. Cells of individual lines were seeded on culture plates/dishes at a density appropriate for a given experiment.

#### 3.1.2. Determination of Cell Viability by Metabolic Microplate Spectrophotometric Assay with MTT

Cell viability was determined using the metabolic microplate spectrophotometric assay with Methylthiazolyl diphenyltetrazolium bromide (MTT), i.e., 3-(4,5-Dimethyl-2-thiazolyl)-2,5-diphenyl-2H-tetrazolium bromide (Sigma-Aldrich; St. Louis, MO, USA). The anticancer activity of the investigated compounds was evaluated in vitro on the basis of their ability to inhibit the proliferation of various cancer cells. The appropriate number of cells for a given cell line (6–8 × 10^3^ cells/mL) were seeded into sterile, colorless 96-well microplates. After 24 h of culture, specific wells of the microplate were treated with test compounds (10–200 µmol/L concentration) and incubated in a CO_2_ incubator for 24 h. The medium was then removed, the cell monolayer was washed twice with sterile PBS solution and fresh culture medium was added to the wells of the microplate.

After 48 h of cell culture, the MTT test was conducted. In each well of the microplate, 50 µL of tetrazolium salt solution (final concentration 0.05 mg/mL) was added. The microplate was then placed in a CO_2_ incubator for 3–4 h (depending on the cell line). Following incubation, 100 µL of DMSO (dimethyl sulfoxide, Sigma-Aldrich, St. Louis, MO, USA) was added to the microplate wells to dissolve the purple formazan crystals formed in the environment of living cells. The absorbance of the formazan solution was read at a wavelength of λ = 580 nm and a reference wavelength of λ = 720 nm after gentle mixing (Biotek Power Wave HT, Biotek Instruments, Winooski, VT, USA).

The percentage of live cells was calculated by comparing the absorbance value of the tested samples with that of the control (cells not treated with compounds). Control absorbance was arbitrarily considered to be 100%. The IC_50_ concentration of the compounds was also calculated, which reduced the percentage of viable cells by 50%. Moreover, cisplatin (Sigma-Aldrich, St. Louis, MO, USA) was used as a reference compound, and identical incubation conditions were used as described above.

#### 3.1.3. Changes in Plasma Membrane Fluidity (Measurement of TMA-DPH and DAUDA Fluorescence Anisotropy)—Lipid Peroxidation

Changes in plasma membrane fluidity were determined by fluorescence spectroscopy and two fluorescent probes located at different depths of the lipid bilayer: TMA-DPH ((1-(4-Trimethylammoniumphenyl)-6-Phenyl-1,3,5-Hexatriene p-Toluene sulfonate, Sigma-Aldrich; St. Louis, MO, USA) and DAUDA (11-[5-(Dimethylamino)-1-naphthalenesulfonylamino] undecanoic acid, Sigma-Aldrich, St. Louis, MO, USA). TMA-DPH is incorporated in the superficial area of the outer monolayer, while DAUDA penetrates into the deeper hydrophobic area of the membrane. The fluorescence of the probes was measured on a Perkin-Elmer spectrofluorometer (Waltham, MA, USA) with an attachment for measuring fluorescence anisotropy with parallel and perpendicular positions of the polarizers.

Cells were seeded on sterile 35 mm dishes in numbers appropriate for a given cell line. After 24 h of culture, the tested compounds were added to the cell culture medium at final concentrations equal to the calculated IC_50_ concentrations. After 24 h incubation, the cells were trypsinized and transferred to Eppendorf tubes (0.5 mL of suspension/sample). The cell suspension was centrifuged at 2500 rpm for 5 min at 4 °C. The cell pellet was washed twice with cold PBS (phosphate-buffered saline) (0–4 °C) and suspended in Tris-HCl/KCl buffer (50 mM Tris-HCl, 0.15 M KCl). The prepared samples were placed on ice. Immediately before measurement, the appropriate fluorescent probe was added to the samples (final concentration of 10^−6^ µmol/L) and incubated on ice in the dark. Fluorescence measurements were made on a Perkin-Elmer LS-5B spectrofluorometer (MA, USA) in an attachment with a polarizer, reading the fluorescence anisotropy at an excitation wavelength λex = 365 nm and emission wavelength λem = 425 nm (TMA-DPH), as well as excitation wavelength λex = 350 nm and emission wavelength λem = 420 nm (DAUDA).

#### 3.1.4. Analysis of the Generation of Reactive Oxygen Species—Measurement of Superoxide Anion

The level of superoxide anion (O_2_^•−^) was determined by microplate spectrofluorimetry with dihydroethidium (hydroethidine, DHE) (Life Technologies, Carlsbad, CA, USA), which is the most commonly used fluorogenic probe for monitoring the intracellular production of superoxide anion O_2_^•−^ [41]. Dihydroethidium displays blue fluorescence in the cytosol, transitioning to bright red upon oxidation to ethidium and subsequent DNA intercalation. Thanks to this shift, DHE can be used as a sensitive probe for cellular ROS. The reaction between superoxide and DHE generates a highly specific red fluorescent product, 2-hydroxyethidium (2-OH-E^+^).

The cells were seeded in black 96-well fluorometric microplates at a density of 10 × 10^3^ cells per well. After 24 h, the cells were incubated with the investigated compounds (IC_50_ concentration) for another 24 h. After incubation, the medium was removed, the cell monolayer in the wells was washed twice with HBSS (140 mM NaCl, 5 mM KCl, 0.8 mM MgCl_2_, 1.8 mM CaCl_2_, 1 mM Na_2_HPO_4_, 10 mM HEPES and 1% glucose) and 50 μL of DHE solution in HBSS was added to each well (final probe concentration 5 μmol/L). Cells were incubated with the probe for 30 min in the dark in an incubator (37 °C), and then fluorescence was measured. The measurement was performed at the following excitation and fluorescence emission wavelengths: λex = 498 nm/λem = 582 nm. Fluorescence measurements were made on a Perkin-Elmer LS-5B spectrofluorometer (MA, USA). Fluorescence intensity values for control cells were arbitrarily set as 100%. The results are presented as percentages of control cell values, calculated relative to the fluorescence intensity of the controls (100%): sample fluorescence was calculated as Δf in relation to the controls.

#### 3.1.5. Analysis of the Generation of Reactive Oxygen Species—Measurement of Hydrogen Peroxide

The study used 2′,7′-dichlorodihydrofluorescein diacetate (H_2_DCFDA, Sigma-Aldrich; St. Louis, MO, USA) for measuring cellular H_2_O_2_; this probe is widely used for this purpose [42]. In the form of a reduced fluorescent probe, it freely diffuses through the cell membrane. Inside the cell, it is hydrolyzed (deacetylated) to non-fluorescent H_2_DCF under the influence of esterases. The H_2_DCF is then oxidized in the presence of generated ROS to form highly fluorescent 2′,7′-dichlorofluorescein (DCF). DCF accumulates in the cytosol, showing characteristic excitation and emission maxima at λex = 498 nm and λem = 522 nm. The fluorescence intensity of DCF is proportional to the concentration of ROS in the cell.

The cells were seeded in black 96-well fluorometric microplates at a density of 10 × 10^3^ cells per well. After 24 h, the cells were incubated with the investigated compounds (IC_50_ concentration) for another 24 h. After this incubation, the medium was removed, the cell monolayer in the wells was washed twice with HBSS (140 mM NaCl, 5 mM KCl, 0.8 mM MgCl_2_, 1.8 mM CaCl_2_, 1 mM Na_2_HPO_4_, 10 mM HEPES and 1% glucose) and 50 μL of H_2_DCFDA solution in HBSS was added to each well (final probe concentration 5 μmol/L). Cells were incubated with the probe for 30 min in the dark in an incubator (37 °C), and then the fluorescence was measured. The measurement was performed at the following excitation and fluorescence emission wavelengths: λex = 498 nm/λem = 522 nm. Fluorescence measurements were made on a Perkin-Elmer LS-5B spectrofluorometer (MA, USA). Fluorescence intensity values for control cells were arbitrarily set as 100%. The results are presented as percentages of control cells calculated on the basis of a comparison of the fluorescence intensity ratio of test samples and controls (100%) (fluorescence of samples was calculated as Δf in relation to the control).

#### 3.1.6. Analysis of the Generation of Reactive Nitrogen Species—Measurement of Nitric Oxide

Diaminofluorescein-FM diacetate (DAF-FM diacetate, Invitrogen, Carlsbad, CA, USA), is a widely employed cell permeant fluorescent reagent for the detection and quantification of reactive nitrogen species (RNS), particularly nitric oxide (NO). In its unmodified state, DAF-FM exhibits minimal fluorescence. However, upon reaction with NO, it undergoes a transformation that yields a highly fluorescent benzotriazole derivative (excitation/emission maxima: λex = 495, λem = 515 nm) [43].

The cells were seeded in black 96-well fluorometric microplates at a density of 10 × 10^3^ cells per well. After 24 h, the cells were incubated with the investigated compounds (IC_50_ concentration) for another 24 h. After incubation with the compounds, the medium was removed, the cell monolayer in the wells was washed twice with HBSS (140 mM NaCl, 5 mM KCl, 0.8 mM MgCl_2_, 1.8 mM CaCl_2_, 1 mM Na_2_HPO_4_, 10 mM HEPES and 1% glucose) and 50 μL of DAF-FM diacetate solution in HBSS was added to each well (final probe concentration 5 μmol/L). Cells were incubated with the probe for 30 min in the dark in an incubator (37 °C). After incubation, the probe was removed and replaced with fresh HBSS buffer and then incubated for an additional 30 min to allow for complete de-esterification of the intra cellular diacetates. After this time, the fluorescence was measured. The measurement was performed at the following excitation and fluorescence emission wavelengths: λex = 495 nm/λem = 515 nm. Fluorescence was measured on a Perkin-Elmer LS-5B spectrofluorometer (MA, USA). Fluorescence intensity values for control cells were arbitrarily set as 100%. The results are presented as percentages of control cells calculated on the basis of a comparison of the fluorescence intensity ratio of test samples and controls (100%) (fluorescence of samples was calculated as Δf in relation to the control).

#### 3.1.7. Assessment of the Contribution of Reactive Oxygen Species to the Cytotoxic Activity of Pyrazoline Derivatives

In order to determine the contribution of ROS to the cytotoxicity of the tested compounds, the cells were pre-incubated for 1 h with the antioxidant vitamin E (Trolox, Sigma-Aldrich; St. Louis, MO, USA). Cells were seeded into sterile, colorless 96-well microplates at a concentration of 6–8 × 10^3^ cells/mL, depending on the cell line. After 24 h of culture, Trolox was added at a final concentration of 50 μM/well. After 1 h of incubation, the antioxidant was washed off, then appropriate concentrations of test compounds (10–200 µmol/L concentration) were added to fresh cell culture medium and incubated for 24 h. The MTT test was then performed under identical experimental conditions, as described in Section 3.1.2.

#### 3.1.8. Analysis of Changes in Transmembrane Mitochondrial Potential (ΔΨm)—JC-1 Method

The mitochondrial transmembrane potential (MMP, ΔΨm) is created by differences in the electrical potential across the inner mitochondrial membrane. Measuring MMP is useful for assessing mitochondrial function and disorders in mitochondrial membrane function, such as depolarization and hyperpolarization. The decrease in MMP may be related to the generation of free radicals and the induction of the internal (mitochondrial) apoptosis pathway [44].

5,5,6,6-Tetrachloro-1,1,3,3-tetraethylbenzimidazolylcarbocyanine iodide (JC-1) is a cationic carbocyanine dye commonly used as a fluorescent probe to detect mitochondrial membrane depolarization. In living cells with an intact mitochondrial membrane, the lipophilic JC-1 penetrates the negatively charged mitochondrial matrix as a monomer. It then forms fluorescent aggregates with red-orange fluorescence (λex = 514 nm, λem = 590 nm). During depolarization of the mitochondrial membrane or in apoptotic cells, the concentration of JC-1 decreases, and the aggregates disintegrate into monomers emitting green fluorescence (λex = 514 nm, λem = 529 nm). The fluorescence ratio of aggregates (λem = 590 nm) and monomers (λem = 529 nm)—590 nm/529 nm—reflects the level of damage to the mitochondrial membrane.

The cells were seeded in black 96-well fluorometric microplates at a density of 10 × 10^3^ cells per well. After 24 h, the cells were incubated with the investigated compounds (IC_50_ concentration) for another 24 h. After incubation with the compounds, the medium was removed, the cell monolayer in the wells was washed twice with HBSS (140 mM NaCl, 5 mM KCl, 0.8 mM MgCl_2_, 1.8 mM CaCl_2_, 1 mM Na_2_HPO_4_, 10 mM HEPES and 1% glucose) and 50 μL of JC-1 solution in HBSS was added to each well (final probe concentration 5 μmol/L). Cells were incubated with the probe for 30 min in the dark in an incubator (37 °C). After incubation, the probe was removed and replaced with fresh HBSS buffer. Fluorescence measurements (Perkin-Elmer LS-5B spectrofluorometer, MA, USA) of JC-1 monomers and aggregates were made at the following excitation and emission wavelengths: λex = 514 nm, λem = 529 nm (monomers), λex = 514 nm, λem = 590 nm (aggregates). The results of the tested samples are presented as the percentage fluorescence ratio of JC-1 aggregates and JC-1 monomers relative to the fluorescence ratio of JC-1 aggregates and JC-1 monomers of the control, which was assumed to be 100%.

A positive control was formed of cells pre-incubated with 5 µmol/L carbonyl cyanide m-chlorophenylhydrazone (CCCP), a chemical inhibitor of oxidative phosphorylation that acts by uncoupling the protons flowing along the electrochemical gradient from phosphorylation.

To visualize changes in mitochondrial potential using the JC-1 probe, microscopic images were captured with a Nikon Eclipse Te200 microscope (Tokyo, Japan) with an Axiocam 208 color microscope camera (ZEISS, Oberkochen, Germany). Cells were seeded at a density of 3 × 10^5^ cells onto sterile 35 mm diameter dishes and cultured for 24 h to allow them to reach the logarithmic growth phase. Following this, the test compounds were added (IC_50_ concentration). The experiment was carried out under the conditions described above and pictures were taken at a magnification of 20× with a Nikon LWD Ph1 DL 20 × 0.40 lens (Tokyo, Japan).

#### 3.1.9. Determination of the Fraction of Apoptotic and Necrotic Cells by Fluorescence Microscopy (Double-Staining of Cells with Hoechst 33258 and Propidium Iodide Fluorescent Dyes)

The simultaneous use of two fluorescent dyes, propidium iodide (PI) and Hoechst 33258, allowed the identification of four cell types in the same sample: viable (weak, dull light blue fluorescence), early apoptotic (bright light blue fluorescence), late apoptotic (pink-purple fluorescence) and necrotic (intense red fluorescence). Propidium iodide (PI) has a negative charge and only penetrates cells with damaged cell membranes, while Hoechst 33258 freely penetrates the intact membrane of living and early apoptotic cells. The cells were seeded at a density of 3 × 10^5^ cells on sterile 35 mm diameter dishes and cultured for 24 h to allow them to reach the logarithmic growth phase. Following this, the test compounds were added (IC_50_ concentration). After 24 h incubation with the compounds, the medium was removed and the cell monolayer in the wells was washed twice with HBSS (140 mM NaCl, 5 mM KCl, 0.8 mM MgCl_2_, 1.8 mM CaCl_2_, 1 mM Na_2_HPO_4_, 10 mM HEPES and 1% glucose). The HBSS solution containing the PI and Hoechst 33258 fluorescent dyes was then added to the cell monolayer at a final concentration of 2 mg/mL each. The cells were incubated with the dyes for 5–7 min in the dark at room temperature. Following this, cell death was analyzed using an Eclipse Te200 microscope (Nikon, Tokyo, Japan). From each dish, 300 cells were counted in the field of view; each variant was tested as three independent replicates. The sum of all cells was taken as 100%, and the content of live, early and late apoptotic and necrotic cells was calculated as a fraction of the total.

In addition, to visualize cancer cell death, microscopic images were captured using an Eclipse Te200 microscope (Nikon, Tokyo, Japan) with an Axiocam 208 color microscope camera (ZEISS, Oberkochen, Germany). The experimental conditions were as described above. Images were taken at 20× magnification using an LWD Ph1 DL 20 × 0.40 lens (Nikon, Tokyo, Japan).

### 3.2. Biocompatibility Assessment

A preliminary assessment of the impact of β-CD, as well as compounds **3** and **5**, and their combinations with cyclodextrin (**β-CD + 3** and **β-CD + 5**), was performed using a previously described biocompatibility assessment model [45]. Briefly, the impact of the studied compounds was determined using red blood cells (RBCs) (erythrotoxicity and RBC morphology) and plasma proteins responsible for the blood coagulation process (determination of APTT, PT and TT times).

#### 3.2.1. Preparation of Solutions of Tested Compounds to Assess Biocompatibility

Solutions of compounds **3** and **5** and their **β-CD + 3**, **β-CD + 5** inclusion complexes were prepared according to the procedure. Following this, 0.685 mg of compound **3** was weighed and dissolved in 2 mL of methanol to obtain a concentration of 1 mmol/L. Similarly, a solution of **5** was prepared by dissolving 0.625 mg in 2 mL of methanol to obtain a concentration of 1 mmol/L. A pure β-CD solution with a concentration of 1 mmol/L (2.269 mg dissolved in 2 mL H_2_O) was prepared. The inclusion complexes were prepared in a 1:1 molar ratio based on physicochemical measurement studies. The concentration of the inclusion complex solution was 1 mmol/L. A solution of the **3** and **5** compounds in methanol was added dropwise to a vial placed on a magnetic stirrer with an aqueous β-CD solution. The solution was stirred for 24 h and then left in the refrigerator to establish equilibrium. The initial solutions of the tested compounds were diluted to obtain the following concentrations: 100 mmol/L, 50 mmol/L, 25 mmol/L, 10 mmol/L and 1 mmol/L.

#### 3.2.2. Red Blood Cells Lysis Assay

The experiments on human blood were performed in accordance with Polish national guidelines, and the study protocols were approved by the Bioethics Committee of the Medical University of Lodz, Poland: approval no (RNN/104/20/KE). The blood samples were obtained from healthy donors at the Blood Donation Centre in Lodz. The procedure for plasma preparation for erythrotoxicity studies is given in a previous paper [17]. The influence of the synthesized compounds, together with a reference compound on RBC membrane integrity, was performed by lysis assay, described elsewhere [17]. Briefly, 2% RBC suspension in 0.9% NaCl was incubated at 37 °C for 24 h with the tested compounds at concentrations of 100 µmol/L, 50 µmol/L, 25 µmol/L, 10 µmol/L and 1 µmol/L. The samples were centrifuged at 3000 rpm for 10 min, and the absorbance of the supernatant was measured at 550 nm. The results are presented as a percentage of hemolysis where a sample containing 10 μL of 2.0% *v*/*v* Triton X-100 was regarded as a positive control contributing to 100% of hemolysis. A sample containing saline solution represented the spontaneous hemolysis of RBCs (control). The experiments were conducted using at least three different biological materials (*n* = 3). The results are presented as mean ± standard deviation (SD). Additionally, erythrocyte morphology was evaluated using a phase contrast Opta-Tech inverted microscope at 400× magnification.

#### 3.2.3. Basic Coagulation Tests: PT, INR, APTT, TT

Basic coagulation parameters were determined using Bio-Ksel reagents (Grudziądz, Poland): APTT reagent, calcium chloride, Bio-Ksel PT plus reagent (tromboplastin and solvent) and thrombin (3.0 UNIH/mL) for TT experiments. The methods were calibrated and the coefficient of variation (PT, APTT, TT experiments) was calculated using a calibrator (Bio-Ksel, Grudziadz, Poland), normal plasma (Bio-Ksel, Grudziądz, Poland) and water for injection (Polpharma, Gdańsk, Poland).

The effects of the tested compounds (**3**, **5** and **β-CD**) and their inclusion complexes (**β-CD + 3** and **β-CD + 5**) on the basic coagulation parameters (i.e., PT, INR, APTT and TT) were determined using a coagulometer (CoagChrom-3003 Bio-Ksel, Grudziadz, Poland), as described elsewhere [46]. The experiments were conducted in multiplicates, and their results are presented as mean ± standard deviation (SD). Control samples consisting of distilled water and methanol were tested. All methods were validated using Bio-Ksel normal plasma, which was dissolved in water for injection (Polpharma, Gdańsk, Poland). The reference values for each test are as follows: PT: 9.7–13.8 s; INR: 0.9–1.2; APTT: 28.2–42.3 s; TT: 11.0–16.5 s for 3.0 UNIH/mL of thrombin.

### 3.3. Studies of Physicochemical Measurements

#### 3.3.1. Isothermal Titration Calorimetry (ITC)

The thermal effects of the interaction of α-cyclodextrin, β-cyclodextrin (BCD) and HP-β-cyclodextrin (HPBCD) with **3**, **3a** and **5** were determined using isothermal titration calorimetry. Measurements were carried out using a MicroCal VP-ITC100 microcalorimeter equipped with a 1400 μL measurement cell. All titrations were carried out under isothermal conditions at 25 degrees C in DMSO solution, which was used as a solvent. The mixing speed during the measurements was 351 rpm. The 0.2 mM solution of the ligand (**3**, **3a** and **5**) located in the measurement cell was titrated with the cyclodextrin solution located in the syringe. Each measurement consisted of 50 injections of cyclodextrin with a volume of 5 uL and a concentration of 5 mM. The interval between individual injections was 300 s. The effect of direct interaction of the ligand with CD was calculated by subtracting the thermal effects of diluting 5 mM cyclodextrin in a pure solvent. The dilution effect of the 0.2 mM ligand solution was negligible. The results obtained were processed in the MicroCall Origin 7 software. Using a single active site model (ITC Tutorial Guide) [47], the stability constant K of the resulting complex, its stoichiometry *n* and thermodynamic parameters that describe the ongoing process were determined, viz., enthalpy (ΔH), Gibbs free energy (ΔG) and entropy (ΔS).

#### 3.3.2. UV–Vis Spectroscopy

Spectrophotometric measurements were performed using a SPECORD 50 single-beam UV–Vis spectrophotometer (Analyst Jena, Thuringia, Germany). The calibration lines of the compounds tested were determined for the concentration range of 3.0 × 10^−6^ M to 1 × 10^−4^ M (Figure 21). The molar extinction coefficients for **3**, **3a**, and **5** were determined experimentally in an aqueous ethanol solution; the relevant maximum wavelengths are presented in Table 9. The influence of α-cyclodextrins, β-cyclodextrins and HP-β-cyclodextrins on the solubility of the tested ligands in water was evaluated in three measurement series. Briefly, excess ligand was introduced into closed Eppendorff tubes containing α-CD, β-CD or HP-β-CD solutions, at concentrations ranging from 1 to 90 mM for α-CD and HP-β-CD and from 1 to 15 mM for β-CD. The test tubes were stored for 10 days at a temperature of 298.15 K until equilibrium was achieved. After this time, the solutions were centrifuged using a Microcentrifuge MPW-55 at 13,000 rpm for five minutes. For testing, a clear solution was taken from above the sediment and diluted appropriately so that the spectrum was within the operating range of the spectrophotometer.

### 3.4. Molecular Docking Calculation

Docking studies were carried out using the AutoDock Vina (version 1_1_2) program [48] as implemented in the Chimera system (version 1.14), which was also used for generating graphical illustration of the results in Figure 20 [49]. Structures of receptors were taken from crystal data and prepared for docking using Antechamber (Ambertools21) [50]. These structures were optimized using RM1 semiempirical parametrization [51] using the Gaussian program [52]. RM1 charges were used for both receptors and ligands. Docking was performed in a cube box with 15 Å side length and an origin located at the center of the space inside of the cyclodextrin receptor. Triplicate docking procedures with perturbed torsional angles in ligands yielded the same scores within 0.1 unit—the best scores are reported. Docking parameters included the highest exhaustiveness (level 8) and number of binding modes (level 10). The maximum energy difference was set to 3 kcal/mol.

## 4. Conclusions and Future Direction

This work represents the initial stage in determining the molecular mechanisms of the anticancer activity of flavanone, chromanone and their spiro-1-pyrazoline derivatives. These compounds undoubtedly have very promising cytotoxic properties against breast and endometrial cancer cells. Three of the five tested derivatives showed a wide spectrum of biological activity, for which the calculated IC_50_ was in low micromolar concentrations. Moreover, the pro-apoptotic properties of the tested conjugates are closely related to their potential to generate ROS in cancer cells. ROS seem to play a crucial role in shaping the overall ability of flavanone, chromanone and their spiro-*1*-pyrazoline derivatives to eliminate cancer cells. The next stage in our research will be to examine the genotoxic properties of the compounds and their interaction with DNA, as well as their influence on the expression of genes/proteins involved in apoptosis and the cell cycle.

The tested compounds did not affect the integrity of the erythrocyte membrane over the examined concentration range and did not contribute to hemolysis. Neither did they induce any pathological changes in the morphology of erythrocytes, since only dyscocytes and echinocytes were recognized. Thus, these compounds can be considered biocompatible towards red blood cells within the tested concentration range.

None of the tested compounds showed any statistically significant impact on the extrinsic and intrinsic coagulation pathway. For compounds **3**, **5** and **β-CD + 5**, only a mild effect on the prolongation of APTT time was observed, which may have a beneficial effect in patients with breast and uterine cancer. It has been observed that in these cases are subject to an increased risk of thrombosis, therefore the mild antithrombotic effect of the compounds is considered beneficial.

The stoichiometric parameter (N) describing the number of ligand molecules bound by the cyclodextrin macromolecule, determined calorimetrically, was close to unity for all tested macrocycles. This indicates the formation of complexes with a stoichiometry of 1 (CD): 1 (tested compounds), in which the chromanone derivative molecules are locked inside the cyclodextrin cavity. In the three tested cyclodextrins, complexation is endothermic (ΔH > 0) and accompanied by a strong increase in the disorder of the reactants (ΔS > 0). The binding process of the ligands by cyclodextrins (α-CD, β-CD and HP-β-CD) is thermodynamically spontaneous (ΔG < 0). The complexation constant (K) determined by isothermal calorimetric titration in all the systems discussed has a value greater than 1000 (K > 1000), suggesting the formation of stable complex connections.

UV-Vis studies indicate that, for all formed complexes, a linear relationship was obtained, with the concentrations of **3**, **3a** and **5** increasing with that of the macrocycle. This further confirms the formation of complexes with a 1:1 stoichiometry. A similar trend can be observed in similar systems containing cyclodextrins [53].

Hence, the cyclodextrin complexes demonstrate promising cytotoxicity, while being harmless to red blood cells at physiological concentrations, and exhibit promising thermodynamics of complexation in the hydrophobic interior of cyclodextrin. As such, the tested flavanone derivatives, and their inclusion complexes with cyclodextrin, offer promise as new therapeutic approaches and merit further evaluation.

## Data Availability

The original contributions presented in the study are included in the article/Appendix A, further inquiries can be directed to the corresponding authors.

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
