# Peer review of "Molecular Pro-Apoptotic Activities of Flavanone Derivatives in Cyclodextrin Complexes: New Implications for Anticancer Therapy"

_ijms, 2024, doi:10.3390/ijms25158488_

Round 1

Reviewer 1 Report

Comments and Suggestions for Authors

In my opinion, the manuscript by Adamus-Grabicka is well written and meets the high scientific standards of IJMS. However, from a technical point of view, the article is sloppy. I ask authors to: (i) justification of table titles; (ii) figure 1 should be numbered 1a, and the current figure 1a should be numbered 1b; (iv) please standardize the font in the text and adapt it to the journal requirements; (v) I recommend preparing a graphical abstract.

Author Response

The response to the Reviewer 1

Reviewer's comments

In my opinion, the manuscript by Adamus-Grabicka is well written and meets the high scientific standards of IJMS.However, from a technical point of view, the article is sloppy. I ask authors to: (i) justification of table titles; (ii) figure 1 should be numbered 1a, and the current figure 1a should be numbered 1b; (iv) please standardize the font in the text and adapt it to the journal requirements; (v) I recommend preparing a graphical abstract.

Response to review

Response 1. We would like to thank the Reviewer for his comment and valuable tips. The manuscript has been adapted to the requirements of the journal and the template. We have corrected punctuation, editorial errors and standardize the font in the text . We have renumbered Figure 1 to 1a and 1a to 1b. All changes and added fragments have been marked in color in track changes mode. Additionally, we have prepared a graphical abstract.

Reviewer 2 Report

Comments and Suggestions for Authors

Author Response

The response to the Reviewer 2

Comments 1. Include a more in-depth mechanistic study or discussion on how flavanone derivatives interact with cellular components to enhance ROS production.

Response 1. Thank you for this valuable tip. According to the reviewer's request, we have added an appropriate paragraph where we discussed possible biological and chemical (structural) mechanisms that may determine the generation of ROS dependent on the tested flavanone derivatives. In paragraph 2.1.6. we have included a discussion on this topic along with additional literature. The changes introduced are marked in green

Comments 2. Figure 20 need clarification, complex between which CD and what ligand?

Response 2. This is clearly indicated in the text and figure legend but additional explicit indication has been added to the legend.

Comments 3. What PDB ids data had the crystals used in the paper?

Response 3. In docking studies we used crystal structures of appropriate cyclodextrins from Crystal Structure Database (Groom CR, Bruno IJ, Lightfoot MP, Ward SC. The Cambridge Structural Database. Acta Crystallogr B Struct Sci Cryst Eng Mater 2016;72:171–9. DOI:10.1107/S2052520616003954)

The refcodes for appropriate cyclodextrins are:

BANXUJ DOI:10.1021/ja00397a021

BCDEXD05 DOI:10.1021/ja00091a014

KOYYUS DOI:10.1016/0008-6215(91)89004-Y

LEDROB DOI:10.1016/0008-6215(93)84243-Y

LEDRUH DOI:10.1016/0008-6215(93)84243-Y

Comments 4. I would recommend the use of a DFT optimization technique and performing the technique in duplicate/triplicate to provide the ligand RMSD value.

Reponse 4. DFT is not an optimization technique. We resorted in this contribution to docking and since the scores are sufficiently different for different complexes there was no need to carry out calculations at higher theory levels. Docking was performed in triplicates starting with different ligand torsional angles but the scores were always the same within 0.1 unit – this info has been added to the methodology part.

Comments 5. Provide a more detailed interpretation of the docking scores and their correlation with the observed biological activities.

Response 5. In the literature (Vukic,M.D. Saudi. Pharm. J. 2020, 28, 136–146; Song S. Mater. Sci. Eng. C 2020,106, 110161and Han B. J. Biosci. Bioeng. 2014, 117, 775–779) the enclosing of the drug by native cyclodextrin macromolecules increased the cytotoxicity against examined tumor cell lines. And the same trend is for mianserin CD studies by S. Belica-Pacha et. al., [doi.org/10.3390/ijms22179419].

Vukic, M.D.; Vukovic, N.L.; Popovic, S.L.; Todorovic, D.V.; Djurdjevic, P.M.; Matic, S.D.; Mitrovic, M.M.; Popovic, A.M.;

Kacaniova, M.M.; Baskic, D.D. Effect of _-cyclodextrin encapsulation on cytotoxic activity of acetylshikonin against HCT-116 and MDA-MB-231 cancer cell lines. Saudi. Pharm. J. 2020, 28, 136–146.

Song, S.; Gao, K.; Niu, R.;Wang, J.; Zhang, J.; Gao, C.; Yang, B.; Liao, X. Inclusion complexes between chrysin and amino-appended-cyclodextrins (ACDs): Binding behavior, water solubility, in vitro antioxidant activity and cytotoxicity. Mater. Sci. Eng. C 2020,106, 110161

Han, B.; Yang, B.; Yang, X.; Zhao, Y.; Liao, X.; Gao, C.; Wang, F.; Jiang, R. Host–guest inclusion system of norathyriol with-cyclodextrin and its derivatives: Preparation, characterization, and anticancer activity. J. Biosci. Bioeng. 2014, 117, 775–779.

Comments 6. What visualization software was used to create the images in Figure 20? Discovery Studio, PyMOL... it needs to be specified.

Response 6. The corresponding info has been added to the methodological part

Comments 7. Provide higher resolution images for Fig 5 and scale bar for Fig 1a, Fig 5, Fig 10.

Response 7. The figures indicated in the review have been corrected and adapted to the editorial requirements.

Comments 8. Expand the discussion on the hemolysis assay results and their implications for the safety profile of the compounds.

Response 8. In response to your comment regarding the expansion of the discussion on the hemolysis assay results, we have made the necessary revisions to our manuscript. The expanded discussion is now included in the text and highlighted in red.

Comments 9. Provide a rationale for selecting these particular cell lines (MCF-7, MDA-MB-231, HCC38, Ishikawa, Hec-1-A).

Response 9. The incidence of estrogen-dependent cancers is increasing worldwide. Women are most often diagnosed with cancer of the breast and reproductive system organs, such as the ovaries, cervix or uterine corpus. The research provided in this scientific work is part of a larger project under the "Excellence Initiative - IDUB Research University". This project is based on examining the biological activity of pyrazoline derivatives condensed with chromanone or flavanone and analyzing the molecular anticancer mechanisms in breast/endometrial cancer cells. The main goal of the project is to show whether the tested derivatives will show different activity in estrogen-dependent and -independent cells, and research in this direction is planned for the further part of the research project and the next publication. Based on existing research, it can be concluded that, apart from currently used chemotherapeutics, natural products such as, among others, flavonoids have pleiotropic, multidirectional effects and constitute possible complementary chemopreventive molecules in the fight against breast cancer with fewer side effects than conventional therapy.

The selection of cell lines for scientific research is crucial to obtain reliable and translatable results. In the case of research on hormone-dependent cancers, the selection of appropriate cell lines is particularly important due to the large diversity of their histological and molecular subtypes. MCF-7, MDA-MB-231, HCC38, Ishikawa and Hec-1-A cell lines were selected for research on breast/endometrial cancer to provide a consistent research model of hormone-dependent cancers. Each of these cell lines has a unique molecular profile, allowing the study of different aspects of breast/endometrial cancer biology. For example, MCF-7 is sensitive to estrogen, while MDA-MB-231 is resistant to estrogen, which causes these five cell lines to show different sensitivity to anticancer drugs, which allows for the study of mechanisms of treatment resistance. Therefore, we believe that the simultaneous use of MCF-7, MDA-MB-231, HCC38, Ishikawa and Hec-1-A lines in research is a valuable tool for research on hormone-dependent cancers.

Comments 10. The reported IC50 values for different cell lines exhibit significant variability without a clear explanation for these differences. Provide a more thorough analysis or hypothesis on why certain cell lines are more sensitive to specific compounds.

Response 10. We fully agree with the reviewer's next suggestion regarding the need to discuss differences in cytotoxic activity and IC50 concentration for individual tested compounds and their antiproliferative activity against various types of cancer. As requested, we included an appropriate paragraph in the article where we attempt to analyze and discuss the mentioned research problem. For the sake of order, we include the added paragraph below and at the same time inform you that in this publication it is included in the section regarding the analysis of cytotoxicity results using the MTT test (section 2.1.1.):

The analysis of cytotoxicity results obtained from the MTT method allows us to notice differences in the biological cytotoxic and antiproliferative activity of the tested 6 Flavanone Derivatives. Interestingly, a certain tendency in biological activity was always maintained for each of the analyzed compounds. Individual compounds were either characterized by very attractive antiproliferative and cytotoxic properties, with an IC50 similar to or even better than the reference compound cisplatin, regardless of the cancer cell line used (compounds 1, 3, 5), or they showed very weak cytotoxic activity (compounds 2, 3a, 4) in the range of 10-200 µM for all 6 cancer lines. The obtained results allow us to hypothesize that the total anticancer activity of individual flavanone derivatives is primarily influenced by the Structure-Activity Relationship (SAR). Changes in the structure of a molecule may lead to changes in its ability to interact with receptors or other biological molecules, which in turn may affect its biological activity.

Interestingly, the most active derivatives in their chemical structure at the C3 position of the benzene ring did not have a p,o-methylophenyl-4,5-dihydro-3H-pyrazole ring attached, but a pi double bond connecting another aromatic benzene ring. At the same time, the attachment at the C3 position of five-membered (pyrazoles) ring with a double bond between nitrogen atoms resulted in a significant reduction in the antiproliferative activity of the analyzed derivatives. It should be noted that such a structure-biological activity relationship was always characteristic of all the tested molecules. Therefore, it seems likely that this structural change may be crucial in the context of overall anticancer activity against the cancer cell lines used in the studies.

Moreover, in relation to our research, there is a clear dependence of the anti-proliferative properties of the tested compounds on their pro-oxidant activity. The most biologically active compounds were able to generate ROS in cancer cells. The issue of the level of ROS in cancer cells is discussed in more detail in section 2.1.6.
